# The Impact of Restaurant Social Media on Environmental Sustainability: An Empirical Study

**Juan Gabriel Martínez-Navalón** [1] **, Vera Gelashvili** [1,*] **and Felipe Debasa** [2]

[1]  Department of Business Economics, Faculty of Social Sciences and Law, Rey Juan Carlos University, Paseo Artilleros s/n, 28032 Madrid, Spain; juangabriel.martinez@urjc.es

[2]  Department of Contemporary History and Actual World, Faculty of Social Sciences and Law, Rey Juan Carlos University, Paseo Artilleros s/n, 28032 Madrid, Spain; felipe.debasa@urjc.es

*  Correspondence: vera.gelashvili@urjc.es; Tel.: +34-91-488-80-35

**Abstract:** Social media is currently a powerful way for companies to keep in touch with their customers and promote new products. The main objective of this research paper is to analyze the impact of social media marketing on environmental sustainability in food and beverage service companies in Spain, taking customer satisfaction into account. The variables, such as customer trust, perceived value, and continuance intention, were also studied to determine whether they influence environmental sustainability. In order to achieve the stated objective, a questionnaire was created and the obtained data was analyzed using a PLS-SEM (partial least squares structural equation modeling) methodology. The main conclusion of this study is an important contribution to the academic literature and is also helpful for restaurant managers for planning communication strategies through social media, since environmental sustainability is one of the main concerns of the public.

**Keywords:** environmental sustainability; social media; restaurants; satisfaction; trust

## 1. Introduction

The emergence of social media has changed the way in which companies communicate with their customers, which has allowed them more direct and faster contact using advertising or marketing strategies [1]. In addition, the cost of using social media for advertising is much lower than traditional methods of publicity [2–6]. Aside from the lower cost of advertising, social media is used to promote transparency, efficiency, and openness by many governments [7] and organizations [4,8]. Through social media, companies promote new products and services, announce offers, easily learn the tastes and preferences of their customers, and relate directly with them [9,10]. Therefore, correct use of social media can help companies to attract new customers and turn interested people into potential customers in a meaningful way. Because of the above, in addition to many other reasons, there is no doubt about the beneficial role of social media in business [2,11–13].

Social media can also serve as a tool to promote sustainability [4,14]. The concept of sustainability has evolved during recent decades, and today it is a popular term that is used in different contexts. The meaning of sustainability is strongly dependent on the context in which it is applied and on whether its use is based on a social, economic, or ecological perspective [15–17]. Most of the interpretations of sustainable development in these three areas agree that a company's policies and actions to achieve economic growth must respect the environment and also be socially equitable [16]. Currently, sustainability issues are considered strategic topics to address to ensure the success and operability of an organization, and already represent a determining factor for investors in assessing the profitability of sustainable enterprises. For this reason, companies are increasingly trying to be sustainable [18–20].

In Spain, many researchers [16,21–25] have studied the sustainability of companies in different sectors, but there are no studies on the impact of social media marketing on the environmental sustainability of companies that operate in the food and beverage services sector. Considering that the food and beverage services sector in Spain presents a great force for reducing the unemployment rate and increasing economic growth, the question is whether the companies that are operating in this sector are sustainable from the point of view of their customers and whether they promote environmental sustainability through social media, since, as we have seen above, social media is key for companies to communicate with their customers.

Bearing all these ideas in mind, the main objectives of this research paper were to analyze the impact of social media in promoting environmental sustainability in food and beverage services, specifically in restaurants, and whether these companies promote sustainability through social media for the conservation and maintenance of the environment. In order to achieve these objectives, a survey was designed and the responses were statistically analyzed. Consequently, the social media marketing of these companies is an explanatory variable related to their environmental sustainability. The main results show that social media marketing used by food and beverage service companies promotes environmental sustainability and shows its importance to their customers. This research is an important contribution to the literature on this subject because there are no empirical studies about the impact of social media marketing on the environmental sustainability of the food and beverage sector in Spain.

The paper is organized as follows. Section 2 contains a review of the literature on social media marketing and sustainability. Section 3 presents the hypotheses for this research project. Section 4 briefly describes the data analysis. Section 5 further discusses the empirical data analysis from the findings. The final sections are devoted to the discussion and conclusion of this study.

## 2. Literature Background

### 2.1. Social Media and Marketing

In the 21st century, social media applications have amassed great power as they have been introduced in all fields of life. Applications such as Facebook, Twitter, YouTube, Instagram, or LinkedIn are becoming essential for millions of people who use social media every day to communicate, relate, or share information and content [26]. However, they are not only used for entertainment, but also in business, culture, and politics, among others [27].

One of the areas where the greatest use of social media has been seen is in marketing [10,28–30]. Nowadays, companies considered this to be the best way to get in touch with potential and existing customers [30], and by doing so, companies have the means to increase revenue and efficiency or reduce costs [9,13]. Several researchers consider that social media has a crucial impact through promotional activities on the customers' awareness and perception [4,10,13,31–33], and as such, companies use this tool as one of their business strategies [9,10,34].

If we compare the cost of traditional marketing with social media marketing, we see that the latter one has lower costs [5,6,35] and is more attractive for companies [36]. Moreover, social media marketing contributes to all phases of the purchasing process, including the consideration and conversation phases and brand recognition [37].

However, there are not only advantages. One study [13] pointed out that while customers who engage via social media interaction are more easily retained and are more likely to upgrade their customer relationships, they also have more service requests. This means that enterprises should keep in mind that if they have more service requests, they should invest more money in this area. Other disadvantages that digital marketing may have are the lack of user trust because of the big number of frauds related to virtual promotions, cash-on-delivery systems, copyright issues, etc. [35]. As such, companies must use digital marketing very carefully because consumers themselves can become product or even company ambassadors.

From here, the conclusion can be drawn that companies must take great care of their social media marketing to keep old customers, win new customers, and be profitable, among other concerns, since in the age of digital marketing, companies have the possibility to have a larger audience than ever before.

### 2.2. Environmental Sustainability of Social Media Marketing

In today's modern age of social media, companies looking to be successful and create profit must enforce sustainability in all areas of the firm. An early definition of sustainability appears in the United Nations General Assembly Report of the World Commission on Environment and Development (1987) and defines it as: "*development that meets the needs of the present without compromising the ability of future generations to meet their own needs.*" Here, three fundamental components to sustainable development are highlighted: environmental protection, balance between economic growth, and social equity for everyone [16].

Analyzing these three sustainability metrics, we can see that economic sustainability refers to a company's contribution to the ongoing viability of the larger economic system. Meanwhile, social sustainability considers the firm's impact on the local communities in which it operates, ranging from corporate philanthropy to providing safe working conditions [17]. Environmental sustainability refers to an approach of different processes of production and engineering indefinitely, a less negative or neutral effect on all environmental systems [38]. This means preserving and protecting the natural resources and developing alternative sources, sustaining consumption levels, developing green products, and promoting the use of biodegradable products. A study carried out by Dangelico and Pujari [39] regarding environmental sustainability has shown that several motivations drive companies to develop green products, like opportunities for risk minimization, preservation of revenues and reputation, or for new business creation.

An important factor related to sustainability is marketing [40], since the customers are the main marketing variables for the company and having all information transferred properly will determine the success or failure of the business [41]. This is why the new concept of "sustainable marketing" has been developed, which is the combination of marketing and sustainability. Fuller [42] defines sustainable marketing as "*the process of planning, implementing, and controlling the development, pricing, promotion, and distribution of products in a manner that satisfies the following three criteria: customer needs are met, organizational goals are attained, and the process is compatible with ecosystem.*" Even so, few research studies in the field of marketing strategies have explored how enterprises communicate sustainability to the public [43].

As we have seen in the previous section, social media is one of the most popular modern marketing tools [9,10,29–31]. Therefore, the relationship between sustainable marketing and social media can be foreseen. At the international level, some researchers have studied the effect of sustainable marketing through social media and their benefits for the companies [14,44]. When considering the case of Spain, we can see that there are not many studies about this topic and even fewer on the food and beverage sector, which is a sector that contributes significantly to the economic growth of the country.

For this reason, the main objective of this research paper is to analyze the impact of the social media marketing in the environmental sustainability in food and beverage service companies in Spain, taking into account customer satisfaction achieved with the company that promotes environmental sustainability through social media.

### 2.3. Restaurants, Social Networks, and Sustainable Marketing

Social networks are a great challenge but also a great opportunity for the companies [45]. One of the sectors where social networks are of vital importance is the food and beverage sector, where restaurants, bars, cafes, and bakeries are included. In 2015, it was discovered that Spain has the highest percentage of bars in the world [46]. In the case of restaurants, it is considered to be one of the most long-standing and traditional sectors of most economies [47]. Over the last few decades, the restaurant sector as a percentage of the total economy in Spain has been undergoing continuous growth [48–50],

which means that its increase has been favorable for the country's economy. In Spain, more than a third of a tourist's expenditure goes directly to food, and consequently to restaurants [46]. This means that it is a sector of great importance for Spain.

Restaurants, like the vast majority of companies, have had to adapt to the technological age and be present on social networks [51]. They consider that an effective Internet presence will lead to better results, either in terms of the number of visits or number of reservations made [47]. Also, the presence of restaurants on social networks is a positive instrument for attracting customers [45,51]. This is because, normally, customers share their moments of gastronomic leisure on their social networks, write opinions about their experience, search for the opinions, rate the services received, and then make the decisions [46,52]. A study carried out by Miranda et al. [45] concluded that social networks like Facebook, Twitter, Google+, etc. can be considered one of the most powerful marketing tools on the Internet.

The marketing of goods, services, information, and ideas via online social media is considered to be social media marketing [53] and it represents a perfect opportunity for companies to strengthen the emotional bond with their customers, increase their sales, and decrease the cost for marketing. Social media marketing provides an ideal opportunity for companies to promote sustainability [44], which is called sustainable marketing. As mentioned in the previous section, sustainable marketing is what motivates companies to adopt sustainable business practices, always aiming to create a better world. For this reason, managers of social network services need to design various complementary services and tools to promote active and convenient information sharing [52] where users can see that the company has implemented effectively sustainable marketing. The next section describes the hypotheses of the research paper and the relation of social network users' trust and satisfaction regarding sustainability.

## 3. Hypothesis Development

As discussed in Section 2, social media is one of the tools that brings together businesses and their consumers and therefore it is a key factor in the development of companies. Through social media, companies promote their marketing, which allows them to have a lower cost and be more efficient [9,13]. Companies use this tool not only to reduce the costs and increase their efficiency, but also to promote their image, corporate social responsibility, and environmental sustainability, among others.

Analyzing the existing literature on the social media influence on sustainability in general, we can see that there are a few studies on this important topic [7,14,44], but there are no studies analyzing the situation of the food and beverage sector and environmental sustainability through social media. For this reason, in this research paper, our main objective was to analyze the impact of social media marketing in the environmental sustainability in food and beverage service companies in Spain, specifically in restaurants, by considering customers' trust, perceived value, continuance intention, and satisfaction achieved with the company that promotes environmental sustainability for the conservation and maintenance of the environment through social networks.

In order to achieve the stated objective, variables such as perceived value, continuity intention, trust, and satisfaction must be analyzed in order to measure environmental sustainability through social media. These four variables have been studied in depth separately [54–58] and jointly [7,59–63] by several researchers. The conclusions of these studies indicated the importance of these variables for the growth of a company's productivity, financial performance, solvency, or short- and long-term risk reduction. This means that if the companies want to be stable and increase the productive capacity or profitability, they have to have customers that trust them and are satisfied with the products/services/politics of the company. Fornell et al.'s [57] study considered a satisfied customer as an economic asset for the company with high returns and low risk. For all that, we can say that for companies, their customers' satisfaction, continuance intention, trust, and perceived value are important factors. At the same time, consumers are increasingly demanding regarding the quality

of products and services and want to know more about the business and political actions of the company [64–67]. Studies along these lines have shown that when consumers choose, they give preference to companies that respect environmental sustainability [68,69]. All this indicates that there can be a positive and direct relationship between the variables described above and environmental sustainability. Each of these variables and their possible relationships are studied below.

A study carried out by Chen and Lin [7] indicates that perceived value is one of the key measures for the marketing of companies. Through perceived value, it is possible to detect consumers' preferences or purchase intentions, taking into account the multiple components of value, the impact of roles and perceptions, and the importance of competition [58]. In order to measure the consumers' perceived value, the five theoretical types of value must be taken into account: functional value (FV), which measures an offering's ability to fulfil its function; social value (SV), which represents the benefits derived through inter-personal/group interactions; emotional value (EMV), which accounts for benefits obtained from an offering's ability to arouse feelings and/or affective states; epistemic value (EPV), which refers to benefits derived through an offering's ability to arouse curiosity, provide novelty, or satisfy a desire for knowledge; and conditional value (CV), which represents the benefits derived in a specific situational context [70]. The questions for measuring perceived value were included in the questionnaire. Preliminary studies show that the perceived value of the customers leads to positive behavior in terms of continuance intention [7,62]. The term continuance intention or repurchase intention is intended as the intention to continue purchasing specified goods or services from the same business, taking into account customers current situation and circumstances [55,71]. One of the advantages of continuance intention is that it reduce costs for businesses, as it is much more expensive to seek new customers than keep old ones [72]. A study undertaken by Shao, Guo, and Ge [62] has shown that perceived value from the company play a more significant role in facilitating customer satisfaction and continuance intention. Another study carried out by Chen and Lin [7] has found that perceived value by users positively and significantly influence the continuance intention. This means that there is a positive relationship between perceived value and continuance intention. From here, the objective of Hypothesis 1 is to see whether the perceived value by customers influences the continuance intention toward the company, which means the use of goods and services of the company in future.

Therefore, the following hypothesis between the perceived value and continuance intention is proposed:

**Hypothesis H1.** *Perceived value by customers has a positive influence on their continuance intention toward the company.*

Many studies in marketing have determined that perceived value is an antecedent to customer's satisfaction [63,70,73]. This is defined as customers' post-consumption evaluation that is dependent on the perceived quality and value [73]; in other words, it is a process of evaluation between what was received and what was expected by consumers [74]. Customer satisfaction has long attracted the interests of many researchers because of its significance in influencing post-purchase behavior [75]. Different researchers have studied the relationship between customer satisfaction and perceived value [63,70,76,77]. According to these studies, there is evidence that perceived value is a significant determinant of satisfaction. Because of this, the objective of Hypothesis 2 is to determine whether perceived value of a business by consumers has a positive relationship with customer satisfaction.

Therefore, the following hypothesis between the perceived value and satisfaction is proposed:

**Hypothesis H2.** *Perceived value has a positive influence on the satisfaction of customers.*

For companies, it is not an easy task to retain the attention of their customers and to keep them satisfied. To do this, they need to improve customer loyalty through maximizing their trust toward the company and its services and goods [78]. Trust is considered as one of the most important variables toward building a stable relationship between the company and its customers [79,80]. A

study conducted in Spain by Ponte, Carvajal-Trujillo, and Escobar-Rodríguez [81] has proved that online purchase intention is influenced by perceived value and trust. Also, this study shows that trust has a positive influence on perceived value. Another study carried out by García [82] has shown the positive relationship between perceived value and trust. For this reason, the objective of Hypothesis 3 was to determine whether perceived value toward a business by consumers has a positive relationship with customer trust generated by the company.

Therefore, the following hypothesis between the perceived value and trust is proposed:

**Hypothesis H3.** *Perceived value has a positive influence on trust generated by the company in the customer.*

Another important variable of this study is customer satisfaction toward the company. Satisfaction is the degree to which goods and services provided by the company meet a customer's expectations; it is a highly personal assessment that is greatly influenced by individual expectations [56]. Therefore, for the business strategy, it is a key factor that differentiates a company from competitors on the market and has a direct influence on the profitability generated by the company [83]. A study conducted by Chiu et al. [84] has shown that the repurchase intention of goods and services is determined by satisfaction. A positive correlation between satisfaction and continuance intention has been found [7,84]. From here, the objective of Hypothesis 4 was to determine whether the satisfaction of the company's customers influences their continuance intention purchase of products and services from that company.

Therefore, the following hypothesis between the satisfaction and continuance intention is proposed:

**Hypothesis H4.** *Satisfaction of the customers has a positive influence on continuance intention.*

As it was mentioned before, environmental sustainability is a key factor for the companies to create green products, be competitive in the market, maximize profits, maintain employee loyalty, and be attractive to their customers [39,85,86]. Nowadays, consumers are increasingly aware of the environmental, social, and economic implications of their consumption habits; for this reason, consumers demand that companies to be aware of their actions since the company's activities have a significant impact on the environment and generally on the society [86]. Research carried out about environmental management in Spain has shown that environmental management practices have positive effects on financial performance, market success, and customer and employee satisfaction [87]. Taking into account the literature analyzed on customer satisfaction and environmental sustainability, it can be seen that there are no studies that analyze the direct relationship of these two variables. With this in mind, the objective of Hypothesis 5 was to determine whether the satisfaction achieved by customers is linked with the environmental sustainability.

Therefore, the following hypothesis between customer satisfaction and environmental sustainability is proposed:

**Hypothesis H5.** *Satisfaction of the customers generated by the company has a positive influence on environmental sustainability.*

The following hypothesis studies the relation between a customer's satisfaction and trust. A study carried out in Spain about travel agencies has proved that the more satisfied a customer is in their relationship with the travel agency, the more trust they place in it [54]. Further studies in marketing has also been found to show that customer satisfaction has a positive impact on customer trust [88,89]. Because of this, the objective of Hypothesis 6 was to analyze whether the satisfaction achieved by customers is related with trust, where if the customers are satisfied, this generates the trust they have for that company and its actions.

Therefore, the following hypothesis between the satisfaction and trust is proposed:

**Hypothesis H6.** *Satisfaction of the customers generated by the company has a positive influence on trust.*

Customer trust has been studied extensively by marketing and management researchers over the last three decades [80,90–93]. Companies that gain the trust of their customers are more likely to expand, grow, and improve their profitability [94]. The relationship between trust and continuance intention has been studied, where the positive relationship between these two variables has been shown [95,96]. From here, the objective of Hypothesis 7 was to determine whether customer trust is positively related to the intention to continue of the customer or user of this company.

Therefore, the following hypothesis between the trust and continuance intention is proposed:

**Hypothesis H7.** *Customer trust has a positive influence on their continuance intention toward the company.*

For companies, building customer trust is a complex process that involves technology and business practices, as well as movement from initial trust formation to continuous trust development [80]. Users who trust companies are more loyal, satisfied, and intend to continue buying the same products and services [78–80,96]. Therefore, trust plays an important role in customers' purchase decisions. In the same way, sustainability contributes to the decisions of customers [68,97] since they are aware of the results of their decisions. Even though the two variables are essential for consumers when making decisions about products or services, there are no studies that show the direct relationship of these two variables. For this reason, the objective of Hypothesis 8 was to determine whether customer trust is positively related to environmental sustainability. This means that if a customer trusts the company and its actions, this will have an impact on sustainability.

Therefore, the following hypothesis between the trust and environmental sustainability is proposed:

**Hypothesis H8.** *Customer trust has a positive influence on environmental sustainability.*

Today, consumers are increasingly informed and have the tools necessary to evaluate the purchase of a product or service, so their decisions regarding whether to continue to buy the same products and services depends on the perceived factors of the company. Studies conducted by Chen [68] and Mendleson and Polonsky [69] have shown that consumers are in favor of companies that practice environmental sustainability. In addition, customers that are satisfied with the activities developed by the company are more willing to continue purchasing from the company [7,62]. In our study, we tested whether companies' social networks users that have the intention to continue purchasing the company's products and services influences environmental sustainability. From here, the objective of Hypothesis 9 was to determine whether there is a positive correlation between customer's intention to continue to use goods and services of the company and environmental sustainability.

Therefore, the following hypothesis between the continuance intention and environmental sustainability is proposed:

**Hypothesis H9.** *Customer's continuance intention in the company has a positive influence on environmental sustainability.*

According to the empirical studies consulted, the proposed hypothesis should be positive. Based on the hypotheses discussed above, we proposed the following research model (Figure 1). The proposed research model for this study is a new model since there have been no researchers that study the relationship of variables such as customer trust, satisfaction, and continuance intention with environmental sustainability. An empirical study on customer experience and perceived value on sustainable social relationship in blogs using the same variables was carried out by Chen and Lin [7]. Therefore, the factors and items of that study have been adapted to build our research model.

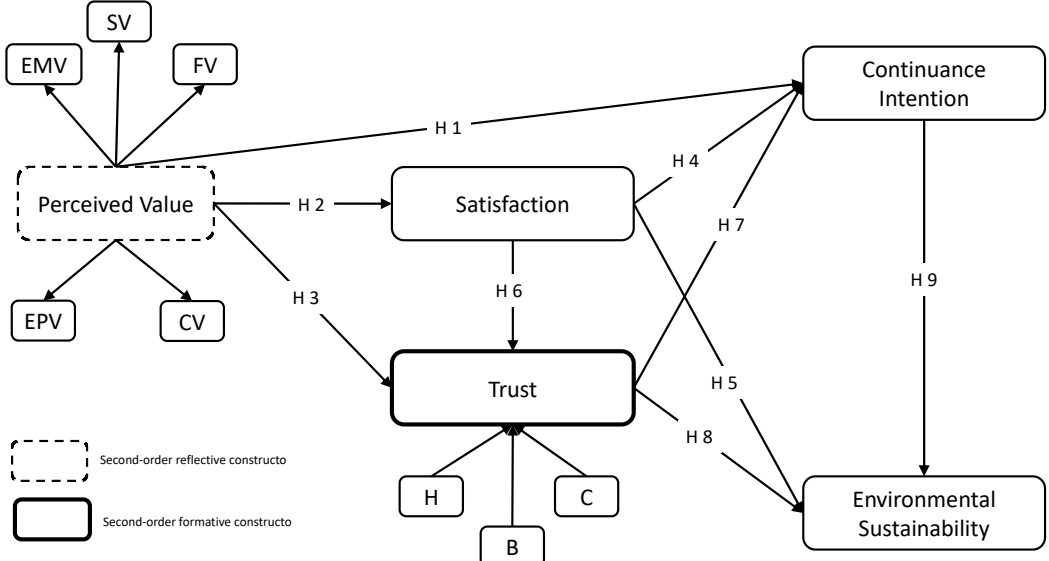

**Figure 1.** Proposed research model. Own elaboration based on Chen and Lin [7]. Emotional Value (EMV); Social Value (SV); Functional Value (FV); Epistemic Value (EPV); Conditional Value (CV); Honesty (H); Benevolence (B); Competence (C).

## 4. Data Analysis

### 4.1. Data

To carry out this study, which analyses the impact of social media marketing on the environmental sustainability in food and beverage service companies, an online questionnaire has been carried out, the purpose of which was to collect a more open and plural sample. A sample of 325 completed questionnaires was collected in June and July 2019. The questionnaire was disseminated through publication on social media (Facebook and Twitter), via e-mail to a convenience sample, and in person. The questionnaires collected in person were 183 of the total sample. The face-to-face questionnaires were obtained from random passers-by in two cities in central Spain (Madrid and Albacete). These two Spanish cities were chosen because they are very different: Madrid is a cosmopolitan city with a very varied population in terms of customs and nationalities, while Albacete is a very traditional city with deep-rooted values and a low percentage in the diversity of nationalities [98]. This difference generates a greater possibility of obtaining a more heterogeneous sample, which makes it possible to obtain a representative sample for the Spanish population.

In the development of the questionnaire, it was taken into account that this questionnaire did not last more than 4 min and that it was composed by two differentiated blocks. The first block involved the classification of the user and the second block investigated the intensity of the feeling of the different questions posed.

A five-point Likert scale questionnaire was used to measure sentiment. The reason for this choice was that Likert-scale questionnaires are the most recommended when carrying out online and face-to-face surveys due to the ease of response on the part of the user and the possibility of collecting the intensity of the sentiment, which allows for a broader study to be carried out [99,100]. The measurement scale had a five levels, from "strongly agree," which equals level 5, to "strongly disagree," which equals level 1.

The sample obtained (Table 1) was composed almost equally of women and men, with 52% being women and 48% being men. The predominant age profile was 31–45 with 55.1%, followed by 18–30 with 19.4% and 46–55 with 17.2%. The sample resided mainly in towns with more than 100,000 inhabitants and the majority were employees (64.8%), followed by self-employed workers with 13.8%. Regarding the level of studies in the sample, university studies accounted for 58.2%, followed

by basic studies with 17.2%. Regarding to the time spent using social media, 38.2% used social media between 30 and 60 min a day, followed by 28% who used them for less than 30 min. With reference to which social media they used, the possibility that respondents could check more than one option was given, highlighting above all that the social media Facebook was used by 295 respondents out of 325, i.e., 90% of respondents used Facebook, followed by Instagram, which was used by 60.9% and Youtube by 55.7%. Finally, we can see how the social media least used by respondents was Snapchat, which was used by only 4.6%.

**Table 1.** Sample Characteristics (*n* = 325).

| Classification Variable | Variable | Frequency | Percentage |
|---|---|---|---|
| Gender | Female | 169 | 52% |
| | Male | 156 | 48% |
| Age | 18–30 | 63 | 19.4% |
| | 31–45 | 179 | 55.1% |
| | 46–55 | 56 | 17.2% |
| | 56–65 | 14 | 4.3% |
| | >65 | 13 | 4% |
| Population of the place of residence | <5000 people | 64 | 19.7% |
| | 5000–20,000 people | 63 | 19.4% |
| | 20,000–100,000 people | 65 | 20% |
| | >100,000 people | 133 | 40.9% |
| Employment situation | Student | 33 | 11.4% |
| | Housewife/man | 5 | 1.7% |
| | Unemployed | 24 | 8.3% |
| | Employee | 188 | 64.8% |
| | Self-employed | 40 | 13.8% |
| Level of education | Without studies | 5 | 1.5% |
| | Basic studies | 56 | 17.2% |
| | High School | 40 | 12.3% |
| | Vocational training | 35 | 10.8% |
| | University studies | 189 | 58.2% |
| Minutes devoted to social media per day | <30 | 91 | 28% |
| | 30–60 | 124 | 38.2% |
| | 60–90 | 36 | 11.1% |
| | 90–120 | 34 | 10.5% |
| | >120 | 40 | 12.3% |
| Social media used (multiple response) | Twitter | 116 | 35.7% |
| | Facebook | 295 | 90.8% |
| | Instagram | 198 | 60.9% |
| | LinkedIn | 140 | 43.1% |
| | Youtube | 181 | 55.7% |
| | Snapchat | 15 | 4.6% |
| | Pinterest | 79 | 24.3% |

## 4.2. Data Analysis

For the data analysis and the validation of the hypotheses, a model for the study of structural equations was chosen, which has its origin in variances (SEM,- Structural Equation Modeling). This model made it possible to carry out a statistical analysis of the relationships that were raised by predicting the dependent variables, which allowed us to calculate and quantify the direct and indirect effects that the variables had on each other [100–102].

Among the different SEM techniques, partial least squares (PLS) was chosen in this research. This choice was based on the fact that SEM techniques are among the most complex regarding factor model analysis and composite models [101], which allows a measurement of latent variables and the estimation of structural models [103–105]. Reinartz, Haenlein, and Henseler [106] conducted a study

where more than thirty investigations were studied in order to analyze why the researchers used PLS-SEM, concluding that it was one of the most complete study methods for this type of work [102].

A total of 326 observations were obtained for this study; since this is not a very high sample, it is advisable to use the PLS-SEM technique. Furthermore, the main reason to use PLS-SEM is that this is a novel study [107] as it studies the influence between a series of variables perceived through social networks [108] and there is no diversity of studies that relate the variables analyzed in this work [109]. In addition, it should be noted that being an exploratory approach, this technique is highly recommended [110,111], along with the possibility of being able to analyze variables composed by dimensions [103], as well as assuming that the proposed model is of a compound type, which indicates that PLS-SEM is one of the most accurate techniques for conducting this analysis [112].

In the research model, it can be seen how most variables were modeled with reflective indicators. However, only "trust" was modeled with formative indicators when it grouped its items in dimensions. The reason for modeling most variables with a reflective character was because they were appreciated as an effect of latent variables [113]; that is to say, the reflective indicators were interchangeable [103]. An increase in the variable was reflected in the increase in the indicators since they were highly correlated. With respect to the "trust" variable that can be found in the second-order model, it its indicators has a formative character because it is a multidimensional construction variable. The formative constructions did not have interchangeable indicators; that is to say, a change in one indicator did not necessarily denote a change in other indicators [113].

To determine the minimum sample size for the model posed in PLS, it is advisable to use the tables presented by Cohen [114,115]. In this study, the minimum size was determined using the G*Power software (created at the Heinrich-Heine-Universität Düsseldorf, Germany) [114,115]. The guidelines of Hair et al. [116] and Reyes-Menendez et al. [115] were followed to carry out this study. The parameters used were: 0.80 for the test power (1 − β test error II) and 0.15 for the effect size (f2) [117]. The result indicated that the minimum sample size was 92 questionnaires; therefore, with a sample obtained of 325, the study could be carried out using PLS-SEM.

Finally, to carry out the analysis using PLS-SEM, SmartPLS 3 software (the latest version of PLS, created at the Technische Universität Hamburg-Harburg, Bönningstedt, Germany) was used as it is one of the most used in the investigations that use this technique [118].

## 5. Analysis of the Results

When analyzing results using PLS-SEM, it should be taken into account that the evaluation is carried out in two steps [103,119]. The first step is a validation study of the measuring instruments (measurement scales), and a second step is the analysis of the relationships between constructs. However, in this study it should be noted that there were multidimensional variables and therefore an intermediate step should be taken in which the dimensions are grouped by creating a second order model, which was later analyzed [119].

### 5.1. Measurement Model

First, due to the multidimensional nature of some variables, the measurement scale of the first-order model was analyzed; in this case, all the items found in this model were reflective, which indicates that the following criteria must be taken into account: individual reliability, composite reliability, convergent validity, and discriminant validity.

The study of individual reliability was carried out taking into account the loads (λ). In this step all items exceeded the threshold of 0.707 established by the criterion of Carmines and Zeller [120]. This criterion ensures that all accepted items represent at least 50% of the variance of the underlying construction [102].

Once the individual reliability criterion had passed, the composite reliability criterion was examined using the following criteria as a model based on Nunnally and Bernstein [121], which sets the minimum threshold of Cronbach's alpha values of 0.70 as an adequate level for "modest" reliability in early research stages, and 0.8 or 0.9 for advanced stages. In this study, all the Cronbach alphas

obtained exceeded 0.9. More recent studies indicate that values above 0.95 may be problematic [122]; because of this, the Dijkstra–Henseler's indicator (rho_A) was also taken into account to provide greater robustness and avoid such problems as it is the only measure of consistent reliability [123] since all the coefficients in this study were higher than 0.7.

Furthermore, the convergent validity analysis that evaluates the average variance extracted (AVE) through the criterion of Fornell and Larcker [124] was used, where this criterion sets the minimum value of the constructs as 0.5 [105]. Table 2 shows that all the constructs analyzed were reliable, and all explained more than 50% of the variance of their own items [103], i.e., all constructs exceeded the minimum values of composite reliability and convergent validity.

**Table 2.** First-order measurement items.

| Constructs | Items | Correlation Loading | CA | CR | rho_A | AVE |
|---|---|---|---|---|---|---|
| Epistemic Value | (EPV2) I'm happy to learn new things from the social media of restaurants. | 0.920 *** | 0.881 | 0.944 | 0.882 | 0.893 |
| | (EPV3) The social media of the restaurants brings me new knowledge. | 0.942 *** | | | | |
| Emotional Value | (EMV1) Getting information from the social media of restaurants is pleasing to me. | 0.903 *** | 0.855 | 0.911 | 0.889 | 0.773 |
| | (EMV2) Getting information from the social media of restaurants makes me feel good. | 0.920 *** | | | | |
| | (EMV3) Getting useful information from social media gives me a sense of personal achievement. | 0.812 *** | | | | |
| Social Value | (SV2) The knowledge I acquire through the social media of restaurants helps me to make a good impression on other people. | 0.964 *** | 0.951 | 0.968 | 0.951 | 0.910 |
| | (SV3) Interacting with restaurant social media makes me feel accepted by others. | 0.944 *** | | | | |
| Functional Value | (FV2) The supply of information from the social media of restaurants makes me feel confident. | 1.000 | 1.000 | 1.000 | 1.00 | 1.000 |
| Conditional Value | (CV2) Social media of restaurants can satisfy me when I have a day off. | 0.909 *** | 0.791 | 0.905 | 0.791 | 0.827 |
| | (CV3) When I have free time, the social media of restaurants entertain me. | 0.910 *** | | | | |
| Satisfaction | (SAT1) I feel satisfied with the knowledge that the social media of restaurants bring me. | 1.000 | 1.000 | 1.000 | 1.00 | 1.000 |
| Honesty | (HON1) The restaurants I follow on social media fulfill what they promise. | 0.913 *** | 0.860 | 0.915 | 0.862 | 0.781 |
| | (HON2) I can trust the promises made by the restaurants I follow on social media. | 0.881 *** | | | | |
| | (HON3) The social media of the restaurants I follow are managed ethically and transparently. | 0.856 *** | | | | |
| Benevolence | (BEN1) The social media of the restaurants I follow offer beneficial advice and recommendations. | 0.945 *** | 0.893 | 0.949 | 0.900 | 0.903 |
| | (BEN3) The restaurants I follow on social media care about the interests and benefits, both present and future, of their stakeholders. | 0.956 *** | | | | |
| Competence | (COM1) The restaurants I follow serve the needs of their customers through social media. | 0.924 *** | 0.904 | 0.940 | 0.908 | 0.839 |
| | (COM2) The social media of the restaurants I follow are competent. | 0.920 *** | | | | |
| | (COM3) The restaurants I follow have a knowledge of their customers that allows them to adapt to their needs. | 0.904 *** | | | | |
| Continuance Intention | (CI2) If I could, I'd spend more time following restaurants on social media. | 1.000 | 1.000 | 1.000 | 1.00 | 1.000 |
| Environmental Sustainability | (SUST1) The restaurants I follow on social media have recycling policies. | 0.761 *** | 0.882 | 0.913 | 0.901 | 0.677 |
| | (SUST2) The social media accounts of the restaurants I follow promote positive environmental ethics among everyone. | 0.854 *** | | | | |
| | (SUST3) The restaurants I follow on social media value and protect the environment. | 0.867 *** | | | | |
| | (SUST4) The social media of the restaurants I follow publish pollution awareness messages. | 0.819 *** | | | | |
| | (SUST5) The social media of the restaurants that I follow defend the diversity of nature, promoting that it is to be valued and protected. | 0.810 *** | | | | |

Note: CA—Cronbach's alpha; CR—composite reliability; AVE—average variance extracted; *** *p*-value < 0.001; n/a = not applicable.

In order to complete the validation of the measuring instruments, the study of discriminant validity must be carried out. This analysis analyzes the amount of variance that a construct captures from its indicators (AVE) and must be greater than the variance that this construct shares with other constructs in the model [124]; that is to say, this analysis measures how different the constructs are from each other [125]. The square AVE roots of each construct must be greater with its items than with the items of other model constructs [124]. Table 3 shows the different correlations that have remained once the measurement scale has been refined according to the criteria set out in the discriminant validity.

**Table 3.** First-order measurement model: discriminant validity (Fornell and Lacker [124]).

| Constructs | BEN | COM | CV | EMV | EPV | FV | HON | CI | SUS | SV | SAT |
|---|---|---|---|---|---|---|---|---|---|---|---|
| BEN | 0.950 | | | | | | | | | | |
| COM | 0.797 | 0.916 | | | | | | | | | |
| CV | 0.469 | 0.420 | 0.909 | | | | | | | | |
| EMV | 0.520 | 0.487 | 0.741 | 0.879 | | | | | | | |
| EPV | 0.680 | 0.634 | 0.625 | 0.734 | 0.945 | | | | | | |
| FV | 0.466 | 0.315 | 0.671 | 0.665 | 0.542 | 1.000 | | | | | |
| HON | 0.720 | 0.790 | 0.594 | 0.625 | 0.753 | 0.431 | 0.884 | | | | |
| CI | 0.384 | 0.443 | 0.643 | 0.701 | 0.526 | 0.403 | 0.560 | 1.000 | | | |
| SUS | 0.610 | 0.626 | 0.366 | 0.345 | 0.463 | 0.248 | 0.569 | 0.442 | 0.823 | | |
| SV | 0.249 | 0.197 | 0.714 | 0.798 | 0.529 | 0.540 | 0.406 | 0.648 | 0.315 | 0.954 | |
| SAT | 0.574 | 0.606 | 0.680 | 0.678 | 0.623 | 0.551 | 0.630 | 0.338 | 0.408 | 0.528 | 1.000 |

Note: BEN—benevolence; COM—competence; CV—conditional value; EMV—emotional value; EPV—epistemic value; FV—functional value; HON—honesty; CI—continuance intention; SUS—environmental sustainability; SV—social value; SAT—satisfaction.

Although, according to the criteria established by Fornell and Larcker [124], the scale of measurement could already be confirmed as valid for the criterion of discriminant validity in this study, the heterotrait–monotrait (HTMT) analysis was also carried out. This helped to confirm more rigorously that all constructs reached discriminant validity and that none of the confidence intervals contained a value of one, suggesting that all variables were empirically different [126] (Table 4).

**Table 4.** First-order measurement model: discriminant validity (heterotrait–monotrait ratio (HTMT)).

| Constructs | BEN | COM | CV | EMV | EPV | FV | HON | CI | SUS | SV | SAT |
|---|---|---|---|---|---|---|---|---|---|---|---|
| BEN | | | | | | | | | | | |
| COM | 0.882 | | | | | | | | | | |
| CV | 0.556 | 0.494 | | | | | | | | | |
| EMV | 0.561 | 0.518 | 0.898 | | | | | | | | |
| EPV | 0.766 | 0.708 | 0.749 | 0.823 | | | | | | | |
| FV | 0.495 | 0.327 | 0.754 | 0.717 | 0.578 | | | | | | |
| HON | 0.821 | 0.897 | 0.718 | 0.717 | 0.867 | 0.466 | | | | | |
| CI | 0.405 | 0.465 | 0.723 | 0.764 | 0.560 | 0.403 | 0.604 | | | | |
| SUS | 0.659 | 0.671 | 0.405 | 0.372 | 0.503 | 0.243 | 0.623 | 0.435 | | | |
| SV | 0.269 | 0.212 | 0.824 | 0.898 | 0.579 | 0.553 | 0.448 | 0.664 | 0.322 | | |
| SAT | 0.606 | 0.635 | 0.764 | 0.722 | 0.662 | 0.551 | 0.678 | 0.338 | 0.415 | 0.542 | |

Note: BEN—benevolence; COM—competence; CV—conditional value; EMV—emotional value; EPV—epistemic value; FV—functional value; HON—honesty; CI—continuance intention; SUS—environmental sustainability; SV—social value; SAT—satisfaction.

Once the measurement scale of the first-order model was validated, the items of each dimension of the multidimensional variables were grouped together in order to validate the second-order model or its equivalent, where those variables that were composed by dimensions group and their items in these dimensions became the indicators of the matrix variable. In this study, the multidimensional variables that performed this action were trust [127–130] and perceiver value [78,131,132]. The analysis was performed by specifying dimensions as latent variables and their measurements as manifest variables.

In this process, multidimensional construction was modeled as a multivariate structural model where dimensions were treated as separate but related constructions [133]. In PLS, components are always estimated. This implies that multidimensional models will always be estimated as aggregates [86].

In this model, unlike the first-order model, analyses must be carried out for the formative variables since the confidence dimensions have a formative character.

In the analysis of individual reliability, composite reliability, and convergent validity, the second-order reflective variable "perceived value (PV)" exceeded all the limits of the analysis, as shown in Table 5. For the individual reliability, all the loads were above 0.7, and for the composite reliability, the same occurred as in the first-order analysis, when the "CR" exceeding 0.9 can cause problems, so we observed the rho_A that indicated there was composite reliability [123].

**Table 5.** Second-order measurement model reflective and formative constructs.

| Constructs | Dimensions | Correlation Loading | Correlation (Weights) | CA | CR | AVE | VIF |
|---|---|---|---|---|---|---|---|
| Perceived Value (PV) | Epistemic Value (EPV) | 0.820 *** | n/a | 0.905 | 0.930 | 0.762 | n/a |
| | Emotional Value (EMV) | 0.930 *** | | | | | |
| | Social Value (SV) | 0.836 *** | | | | | |
| | Conditional Value (CV) | 0.883 *** | | | | | |
| | Functional Value (FV) | 0.784 *** | | | | | |
| Trust | Honesty (HON) | 0.970 | 0.668 *** | n/a | n/a | n/a | 2.827 |
| | Benevolence (BEN) | 0.860 | 0.270 *** | | | | 2.917 |
| | Competence (COM) | 0.879 | 0.136 | | | | 3.736 |

Note: CA—Cronbach's alpha; CR—composite reliability; AVE—average variance extracted; VIF—variance inflation factor. *** *p*-value < 0.001; n/a—not applicable.

Similarly, in the discriminant validity analysis of the second-order model, this also complied with the limits established in the Fornell–Larcker and heterotrait–monotrait ratio (HTMT) criteria, as seen in Table 6.

**Table 6.** Second-order measurement model reflective construct: discriminant validity.

| Constructs | Fornell-Larcker Criterion | | | | | Heterotrait-Monotrait Ratio (HTMT) | | | |
|---|---|---|---|---|---|---|---|---|---|
| | TRUST | CI | SAT | SUS | PV | CI | SAT | SUS | PV |
| Trust | | | | | | | | | |
| CI | 0.538 | 1.000 | | | | | | | |
| SAT | 0.658 | 0.338 | 1.000 | | | 0.338 | | | |
| SUS | 0.632 | 0.444 | 0.410 | 0.823 | | 0.435 | 0.415 | | |
| PV | 0.672 | 0.693 | 0.724 | 0.416 | 0.852 | 0.721 | 0.756 | 0.422 | |

Note: CI—continuance intention; SUS—environmental sustainability; PV—perceived value; SAT—satisfaction.

Once the reflective variables were validated, the formative variable "confidence" was validated by applying different criteria. First, problems of collinearity should be avoided [103], such that an evaluation of the variance inflation factor (VIF) is carried out using the criterion of Hair et al. [110]. In this criterion, the dimensions of benevolence and honesty were below 3 and competition was well below 5, which indicated that there were no problems of collinearity. We also used this criterion for the formative variable. Furthermore, the magnitudes of the variable's weights were analyzed [134], as and analysis of its significance was carried out [119]. It can be seen how the significance of the weight of "competence" was not significant but having a load greater than 0.5 was retained, as indicated in Hair et al. [119] (Table 5).

Once the analysis of second-order measuring instruments was carried out, it should be noted that all items were retained.

## 5.2. Structural Model Analysis

Once the scale of measurement was validated, the results of the structural model were analyzed, obtaining the predictive capacities of the model and the relationships that were obtained by means of the hypotheses between variables. The analysis performed on the proposed model was carried out taking into account the evaluation of collinearity by means of the algebraic sign and the significance and magnitude of the coefficients analyzed: path coefficients ($\beta$), the $R^2$ values (variance explained), the effect size $f^2$, and the test $Q^2$ (validated cross redundancy), in order to measure the predictive relevance of the model [103,135,136].

In order to be able to carry out the measurement of magnitudes and meanings, it must be previously verified that there is no multicollinearity between the antecedent variables of each of the endogenous variables [137] by means of VIF values of the model, which must not be higher than 5 [100]. All the VIF values of the model variables were below 2.446, so there was a low degree of multicollinearity [103].

For the analysis of predictive power that indicates the amount of variance of a variable that is explained by another predictive variable (determination coefficient $R^2$), the cut-off points for its analysis were 0.75—relevant, 0.50—moderate, and 0.25—weak [105,110]. It was observed that the values obtained for the trust variables, continuance intention, and satisfaction were moderate; in other words, the results obtained indicate that the model explained 51.3% ($R^2 = 0.513$) of the trust, 57.2% ($R^2 = 0.72$) of the continuance intention, and 52.4% ($R^2 = 0.524$) of the satisfaction. While the value obtained for the variable environmental sustainability was within the cut-off weak points, they were very close to moderate and were able to be considered as having a moderate predictive character by explaining 41.4% ($R^2 = 0.414$).

Complementary to the analysis of $R^2$, the size of the effect $f^2$ was studied, which analyzes the level at which an exogenous variable contributes toward explaining an endogenous variable. The results in Table 7 show that the relationship of the perceived value with the continuance intention and satisfaction variables was large; the moderate relationships were those of trust with environmental sustainability, satisfaction with continuance intention, and perceived value with trust; while the rest were weak with the exception of the satisfaction with environmental sustainability, which lacked an effect.

**Table 7.** Comparison of hypotheses.

| | Path Coeff ($\beta$) | Statistics $t$ ($\beta$/STDEV) | $f^2$ | Confidence Interval | |
| --- | --- | --- | --- | --- | --- |
| | | | | 5.0% | 95.0% |
| H1. Perceived Value → Continuance Intention | 0.831 *** | 16.199 | 0.660 | 0.744 | 0.913 |
| H2. Perceived Value → Satisfaction | 0.724 *** | 21.076 | 1.101 | 0.664 | 0.777 |
| H3. Perceived Value → Trust | 0.410 *** | 6.509 | 0.164 | 0.304 | 0.512 |
| H4. Satisfaction → Continuance Intention | −0.443 *** | 8.564 | 0.193 | -0.528 | −0.359 |
| H5. Satisfaction → Environmental Sustainability | −0.006 (n/s) | 0.089 | 0.000 | -0.108 | 0.100 |
| H6. Satisfaction → Trust | 0.362 *** | 6.378 | 0.128 | 0.267 | 0.453 |
| H7. Trust → Continuance Intention | 0.271 *** | 4.710 | 0.084 | 0.178 | 0.367 |
| H8. Trust → Environmental Sustainability | 0.557 *** | 8.157 | 0.241 | 0.440 | 0.666 |
| H9. Continuance Intention → Environmental Sustainability | 0.146 ** | 2.166 | 0.026 | 0.036 | 0.255 |

$R^2$: Trust = 0.513; Continuance intention = 0.572; Satisfaction = 0.524; Environmental sustainability = 0.414. Adjusted $R^2$: Trust = 0.510; Continuance intention = 0.568; Satisfaction = 0.522; Environmental sustainability = 0.409. $Q^2$: Trust = 0.387; Continuance intention = 0.568; Satisfaction = 0.513; Environmental sustainability = 0.245. ** *p*-value < 0.01, *** *p*-value < 0.001.

Regarding redundancy indices with cross validation ($Q^2$), which serve to examine the predictive relevance of the theoretical/structural model [134], it was observed that since the $Q^2$ indexes were greater than zero, the model had satisfactory predictive relevance.

Regarding the contrast between hypothesis, it was concluded that the relationships were significant, except for the relationship between satisfaction and environmental sustainability, which did not reach

the minimum value of $t = 1.645$, which indicates that it should be rejected; in order to have a greater reliability in the hypothesis contrast, the analysis of confidence intervals (non-parametric analysis) was carried out where it also indicated that it should be rejected. Finally, the satisfaction and continuance intention hypothesis must also be rejected since it had a negative sign for the path coefficient, which indicates that the relationship did not occur in the direction stated in the model. Therefore, the rejected relationships in the model were H4 and H5, and H1, H2, H3, H6, H7, H8, and H9 were accepted, as can be seen in Figure 2.

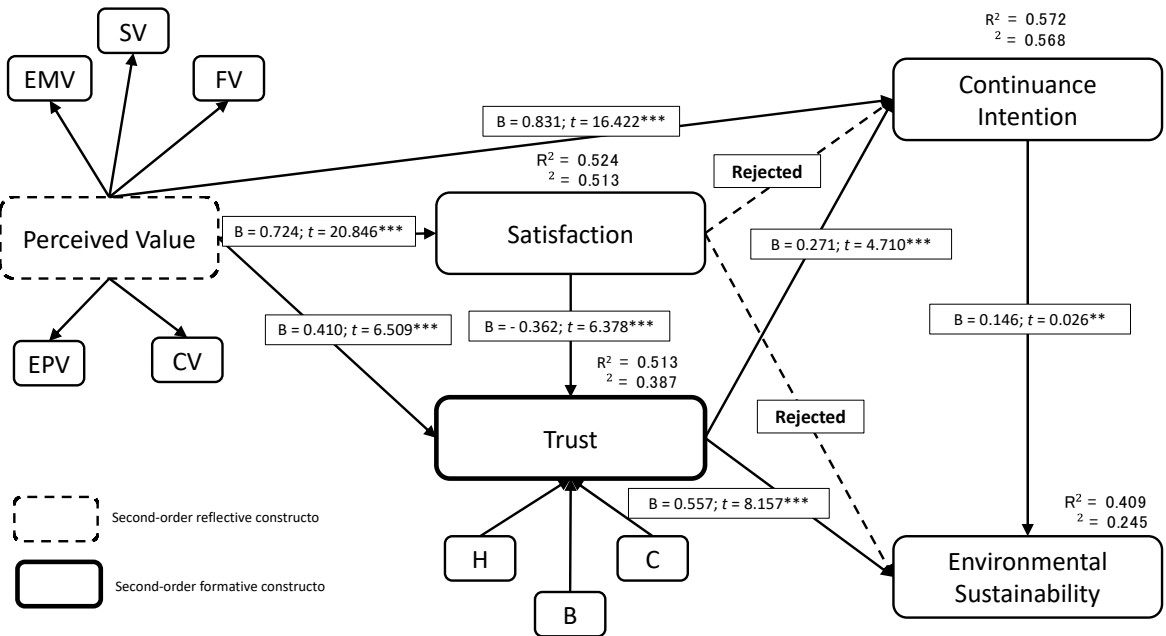

**Figure 2.** Proposed research model.

## 6. Importance–Performance Map Analysis

Finally, once the proposed model was analyzed, the importance–performance map analysis (IPMA) was carried out (Table 8). This analysis can enrich the PLS-SEM analysis by obtaining additional results and findings. This study not only considered the route coefficients, but also considered the average value of the latent variables and their indicators, which shows the performance dimension [138,139]. These novel findings through IPMA provide additional conclusions for management actions as it combines the importance analysis and dimensions of performance in practical PLS-SEM applications [138]. The analysis allows for the identification of the most important areas of specific actions [140]. The results obtained are represented graphically by contrasting the total effects on the *x*-axis with the latent variable scores, reflected on a scale from 0 to 100, on the *y*-axis. The greater the factor yield, the closer the factor is to 100, and all total effects greater than 0.10 are significant when $p \leq 0.05$ [138–140]. Once the target variable (environmental sustainability) was defined, the results were obtained.

**Table 8.** IPMA result of the target construct environmental sustainability.

| Constructs | Importance | Performance |
|---|---|---|
| Perceived Value | 0.360 | 53.242 |
| Satisfaction | 0.109 | 61.077 |
| Trust | 0.419 | 59.775 |
| Continuance Intention | 0.101 | 46.846 |

Once the IPMA was calculated with respect to environmental sustainability, as seen in Table 8 and Figure 3, the yields and effects of the study variables could be appreciated. With respect to

the importance of constructs versus the target variable, it can be seen how trust had a relatively high importance followed by perceived value, and the other two variables had a lower importance. However, the performance of all four variables had very close values, being approximately in the middle, which indicates that they have possibilities for improvement.

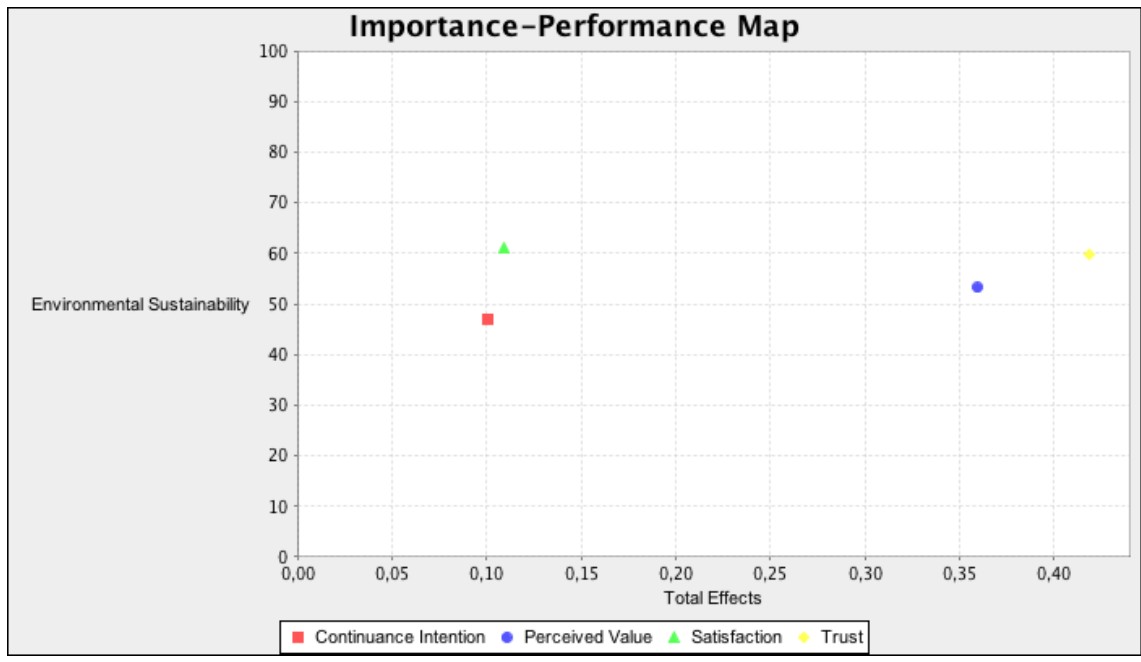

**Figure 3.** IPMA of the target construct environmental sustainability.

## 7. Discussion

The results obtained in the study on social media in restaurants show the direct and indirect influence exerted by variables, such as trust, perceived value, or continuity intention, on environmental sustainability. It should be noted that few studies have analyzed the influence that these variables have on environmental sustainability, and specifically, they are practically non-existent when they focus of the study of these variables is the service sector, as well as on studies that measure these relationships in social media. After a revision of the literature, one of the main problems was to develop this study., because of the scarcity of the academic works on this topic.

Variables as important as perceived value, continuance intention, or satisfaction, which serve to measure the level of influence the organization exerts on its interest groups, have received very little analyses in terms of one of the major communication channels, namely social media. For this reason, it is important to point out that although direct and positive hypotheses have been established in this study, these have been proposed with an exploratory character, which allows us to see whether there is an influence of the proposed variables on environmental sustainability, taking into account the influence that publications in restaurants' social media have had.

The results obtained show very relevant data that should be taken into account for communication strategies regarding the social media of restaurants. First of all, it can be seen that the majority of users surveyed were between 31–45 years old, where a large part has undertaken university studies and have social profiles on Facebook and Instagram.

As for the relationships studied, the perceived value by followers through social media has a direct and positive influence on the continuance intention toward following the restaurant in its social media. This relationship has been extensively studied [7,62] and has a very high influence. Likewise, the influence of perceived value on satisfaction and trust, two of the most important variables of relational marketing, is also direct and positive [141], and these influences were shown clearly and meaningfully.

In the analysis of user satisfaction and their influences, it should be noted that two of the three relationships studied were rejected. It was not demonstrated that the influence of user satisfaction on environmental sustainability was significant; on the contrary, it was demonstrated that their satisfaction has a direct and positive influence on the confidence they have in the restaurant, supporting the majority of the studies analyzed in the bibliography [88,89]. One of the results to emphasize within the research is the non-direct and positive influence of the satisfaction on the intention of continuity, although the relation, if it is significant, the analysis shows that the influence was the inverse one, where the continuance intention influences the satisfaction of the users.

Regarding trust, the two relationships raised were confirmed, where trust directly and positively influenced the intention of continuity, and trust significantly directly and positively influenced environmental sustainability.

Finally, continuance intention also directly and positively influenced environmental sustainability.

In this study, it was found that the influence of the company on users was relevant and that organizations could significantly influence their users such that they had a greater awareness of environmental sustainability through publications on their social media, where Facebook and Instagram were the platforms used most by users.

Another relevant finding were the results obtained from the environmental sustainability and satisfaction relationship, where the results indicated that there was no direct and positive relationship since the relationship was not significant.

## 8. Conclusions

The importance of this study focuses on the relevance of environmental sustainability today for society as a whole and specifically for companies. For this reason, this analysis focused on the services sector, one of the most important sectors for Spain, and specifically on the food and beverage sector. The food and beverage sector is the most accessible to all types of users and the one with the greatest possibility of influencing society. Therefore, it was studied whether restaurants through their social media influenced their followers and thus improved environmental sustainability.

In this study, several conclusions were reached that helped the decision-making of restaurant managers and community managers to plan communication strategies through social media since environmental sustainability has become one of the main concerns of the population.

One of the main objectives that organizations have in their strategies is to keep their followers. This study concludes that a follower who had a high level of perceived value will almost entirely intend to continue having a relationship with the organization. Similarly, a follower who appreciated the value in restaurant social media posts, a follower who is satisfied will build confidence in the company. This means that if an organization wants to increase the levels of satisfaction, trust, and continuance intention, it must generate publications that create value for its followers. Therefore, Hypotheses H1, H2, and H3 were validated.

In terms of satisfaction, it can be seen in the study that a restaurant that had satisfied its followers using social media will generate trust in the followers toward it but will not influence environmental sustainability directly. It was also concluded in this study that satisfaction did not influence the continuance intention. Therefore, Hypotheses H4 and H5 were not validated and Hypothesis H6 was accepted.

Regarding trust, it should be noted that users who had optimal levels of trust in the restaurant were followers who intended to continue following the restaurant, just as a follower with trust was a follower who would positively influence the restaurant's environmental sustainability campaigns. Hypotheses H7 and H8 were therefore validated.

Finally, a follower who intended to continue following the restaurant on social media was a follower who to some extent saw positive environmental sustainability campaigns and collaborated with them. Hypothesis H9 was therefore validated.

It should be noted that if the company wants to carry out environmental sustainability campaigns through the social media of restaurants, it is important that managers enhance the confidence of followers toward the restaurant as this will have a decisive influence on the success of the campaign.

This study has some main implications for food and beverage company managers: First, managers of these companies must pay more attention to the values that are passed on to their customers since there was a positive relationship between customers' perceived value from the company and the variables like continuance intention, trust, and satisfaction. Therefore, if companies had customers who trusted them, were satisfied with the service and goods, and planned to continue buying products and services from the company, this will lower the possibility that the company will have problems in the future. Second, if managers of food and beverage companies want to carry out environmental sustainability campaigns through the social media of restaurants, it is important to enhance the confidence of followers toward the restaurant as this will have a decisive influence on the success of the campaign. Finally, managers of food and beverage companies should know that to be sustainable is the duty of all citizens and if they participate to make their consumers aware of it, they will participate and protect the environment for future generations.

However, these implications cannot be generalized since previous studies have shown that there are different cultural differences that influence on customers' behaviors, motivation, emotion, or evaluation choice among others [142–144], and therefore, the results can vary depending on the country.

As in all studies, this paper is not short of limitations. Our main limitations are in the size, sample, and number of studies consulted previously. Another of the limitations could be the methodology used for this study. According to Hair et al. [110], there are disadvantages when using the PLS-SEM technique. One of the disadvantages to using PLS-SEM is that it focuses on maximizing partial model structures, which means that it first optimizes the measurement model parameters and then, in a second step, estimates the path coefficients in the structural model. Another limitation to use PLS-SEM for theory testing and confirmation is that there is no adequate global measure of the goodness of a model fit. Finally, these researchers point out that the PLS-SEM parameter estimates are not optimal regarding bias and consistency. Despite the limitations, our study can be considered a first step in the study of restaurant social media on environmental sustainability.

Related to future research on this topic, it would be interesting to analyze the impact of social networks on environmental sustainability using samples collected from all regions of Spain. Also comparison of the result of Spain to one or several countries taking into account the variables of cultural differences would also be useful. In future studies, we will try to study different sectors, not only the food and beverage sector, as well as modifying the background variables of environmental sustainability. In addition, it would be interesting to use a different methodology, such as the Q methodology; with this tool we could quantify qualitative answers for customers and it will allow us to detect blind spots.

**Author Contributions:** Conceiving and designing review: J.G.M.-N., V.G., and F.D.; performing the methodology: J.G.M.-N.; analyzing results: J.G.M.-N.; and writing the paper, J.G.M.-N., V.G., and F.D.

**Funding:** This research received no external funding.

**Acknowledgments:** The authors would like to express their sincere gratitude to George Luke Akhalaya, Henry Clark, the anonymous reviewers, and the handling editor for their truly valuable comments.

**Conflicts of Interest:** The authors declare no conflict of interest.

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
