# Peer review of "The Impact of Restaurant Social Media on Environmental Sustainability: An Empirical Study"

_sustainability, doi:10.3390/su11216105_

Round 1

Reviewer 1 Report

The paper analyses the impact of social media over the environmental sustainability. I found the paper to be well written, providing a comprehensive literature review on both social media and structural equations modelling. The hypothesis stated in the paper are well described and in relation with the literature. The questionnaire extracted data are carefully analyzed and presented. The concluding remarks are well written.

Observations:

In the revised version of the paper please add the questions included in the questionnaire as it is hard to understand the connection between each question and the latent component; please revise all the references in rows 154 - 175 as they do not match the required format by MDPI.

Author Response

Dear Reviewer 1,

We are pleased to resubmit for publication the revised version of the manuscript “The impact of restaurant social media on environmental sustainability: An empirical study”. First, we would like to thank you for the opportunity to revise and to resubmit our work.

We have carefully taken your comments and insights on board doing our best in the revised version. Therefore, below we will respond to your comments, we also attach the new version of our study, indicating in pale blue the modifications made.

The paper analyses the impact of social media over the environmental sustainability. I found the paper to be well written, providing a comprehensive literature review on both social media and structural equations modelling. The hypothesis stated in the paper are well described and in relation with the literature. The questionnaire extracted data are carefully analyzed and presented. The concluding remarks are well written.

The Authors

Thanks for your consideration, we are very grateful for the comment. We would really appreciate the opportunity to publish in such a well-known journal.

Observations: In the revised version of the paper please add the questions included in the questionnaire as it is hard to understand the connection between each question and the latent component.

The Authors

We are sincerely grateful for the comment, all the questions used in the study are analysed and put in tables 1 and 2. If the reviewer considers that the tables are not clear and it is difficult to understand the questions, we will redesign the tables.

Please revise all the references in rows 154 - 175 as they do not match the required format by MDPI.

The Authors

Thanks for the comment, we are sorry about the mistake. We have checked and modified the errors.

Reviewer 2 Report

First of all, I want to say to the authors that this work present interesting ideas that are potentially of interest to both academics and practitioners. And this is well designed, carefully conducted empirical research for the field of Sustainability Journal0.

I find this work interesting and honest. But I would like to suggest some ideas to further improve the manuscript. The included comments must be seen as recommendations to improve the quality of the presented work.

This paper seeks to analyze impact of the social media marketing in the environmental sustainability in food and beverage service companies in Spain.

The introduction of the paper delves adequately into the specific research methodology and sample issues. The review of literature it seems so enough and updated. But I consider it is necessary to include a new section related with restaurants, social networks and sustainable marketing. For instance, you can check the follow references between others:

Daries, N.; Cristobal-Fransi, E.; Ferrer-Rosell, B.; Marine-Roig, E. (2018). Maturity and development of restaurant websites: a comparison of Michelin-starred restaurants in France, Italy and Spain. International Journal of Hospitality Management, 73, 125-137. Kang, J. W., Namkung, Y. (2016). Restaurant Information Sharing on Social Networking Sites Do Network Externalities Matter? Journal of Hospitality & Tourism Research, 40(6), 739-763. Miranda, F. J., Rubio, S., Chamorro, A. (2015). The Web as a Marketing Tool in the Spanish Foodservice Industry: Evaluating the Websites of Spain’s Top Restaurants. Journal of Foodservice Business Research, 18(2), 146-162. Rosalin, G., Poulston, J., Goodsir, W. (2016). Strategy communication in family owned restaurants: ad hoc and ad-lib. Journal of Hospitality and Tourism Administration, 17(2), 101-122.

The research questions are not well developed. It would be necessary to clarify the objectives and argued. A case study is developed (but is so local) and it is focused on the peculiarities on the one specific country, Spain. It would be interesting analyse if cultural differences can affect the results.

The methodology used can be deemed appropriate, and the authors’ presentation of the results is clear and concise, thus facilitating the reader’s understanding, but it would be necessary to deepen in the limitations of the chosen methodology. It would be interesting to know whether the results presented differ from previous studies. This would add value to the findings.

Finally, in the discussion and conclusion section it would be interesting deep in managerial implications and future research lines.

Author Response

Dear Reviewer 2,

We are pleased to resubmit for publication the revised version of the manuscript “The impact of restaurant social media on environmental sustainability: An empirical study”. First, we would like to thank you for the opportunity to revise and to resubmit our work.

We have carefully taken your comments and insights on board doing our best in the revised version. Therefore, below we will respond to your comments, we also attach the new version of our study, indicating in pale blue the modifications made.

First of all, I want to say to the authors that this work present interesting ideas that are potentially of interest to both academics and practitioners. And this is well designed, carefully conducted empirical research for the field of Sustainability Journal. I find this work interesting and honest. But I would like to suggest some ideas to further improve the manuscript. The included comments must be seen as recommendations to improve the quality of the presented work.

The Authors

Thanks for your consideration, we are very grateful for the comment. We would really appreciate the opportunity to publish in such a well-known journal.

This paper seeks to analyze impact of the social media marketing in the environmental sustainability in food and beverage service companies in Spain. The introduction of the paper delves adequately into the specific research methodology and sample issues. The review of literature it seems so enough and updated. But I consider it is necessary to include a new section related with restaurants, social networks and sustainable marketing. For instance, you can check the follow references between others:

Daries, N.; Cristobal-Fransi, E.; Ferrer-Rosell, B.; Marine-Roig, E. (2018). Maturity and development of restaurant websites: a comparison of Michelin-starred restaurants in France, Italy and Spain. International Journal of Hospitality Management, 73, 125-137.

Kang, J. W., Namkung, Y. (2016). Restaurant Information Sharing on Social Networking Sites Do Network Externalities Matter? Journal of Hospitality & Tourism Research, 40(6), 739-763.

Miranda, F. J., Rubio, S., Chamorro, A. (2015). The Web as a Marketing Tool in the Spanish Foodservice Industry: Evaluating the Websites of Spain’s Top Restaurants. Journal of Foodservice Business Research, 18(2), 146-162.

Rosalin, G., Poulston, J., Goodsir, W. (2016). Strategy communication in family owned restaurants: ad hoc and ad-lib. Journal of Hospitality and Tourism Administration, 17(2), 101-122.

The Authors

We really appreciate these comprehensive comments. We have found very interesting and useful the papers suggested by the reviewer. Following these papers, we have add the new section “2.3. Restaurants, social networks and sustainable marketing”. Therefore, we have added the following remarks:

- Pages 3-4: “2.3. Restaurants, social networks and sustainable marketing

Social networks are a great challenge but also a great opportunity for the companies [51]. One of the sectors where social networks are of vital importance is food and beverage sector, where restaurants, bars, cafes, bakeries are included. In 2015, Spain has turned out to be the country in the world with the highest percentage of bars per inhabitant [45]. In case of restaurants, it is considered as one of the most long-standing and traditional sector of most economies [46]. Over the last decades, the restaurant sector as a percentage of the total economy in Spain has been in continuous growth [47]–[49], which means that its increase has favored the country's economy. In Spain, more than a third of a tourist's expenditure when it comes goes directly to food, and consequently to restaurants [45]. This means that it is a sector of great importance for Spain.

Restaurants, like the rest of the companies have had to adapt to the technological age and be present on social networks [50]. It is considering that an effective Internet presence will lead to better results, either in number of visits or number of reservations made [46]. Also, presence of restaurants in the social networks is a positive instrument for attracting customers [50], [51]. This is because, normally, customers share their moments of gastronomic leisure on their social networks, write opinions about their experience, search for the opinions, rate the services received and then make the decisions [45], [52]. Study carried out by Miranda et al. [51] has come to the conclusion that social networks like Facebook, Twitter, Google +, etc., can be considered as one of the most powerful marketing tools on the Internet.

Marketing of goods, services, information and ideas via the online social media is considered as social media marketing [53] and it represents a perfect opportunity for companies to strengthen the emotional bond with their customers, increase their sales or decrease the cost for marketing. Social media marketing provides an ideal opportunity for companies to promote sustainability [44] that is called sustainable marketing. As mentioned in the previous section, sustainable marketing is what motivates companies to adopt sustainable business practices, always aiming to create a better world. For this reason managers of social network services need to design various complementary services and tools to promote active and convenient information sharing [52], where users can see that company implemented effectively sustainable marketing. Next section describes the hypotheses of the research paper and relation of social network users trust and satisfaction with sustainability.”

The research questions are not well developed. It would be necessary to clarify the objectives and argued.

The Authors

Thanks for comment, we have extended the explanations of our main research objective in the section “3. Hypothesis development”. We also explain all Hypothesis objectives and rewrote the hypothesis to make them clearer to the readers.

Therefore, we have added the following remarks:

-Page 4: “For this reason, in this research paper, our main objective is to analyse impact of the social media marketing in the environmental sustainability in food and beverage service companies in Spain, specifically in restaurants, considering customers trust, perceived value, continuance intention and satisfaction achieved with the company that promotes environmental sustainability for the conservation and maintenance of the environment through social networks.

Page 5: ….“From here, the objective of hypothesis 1 is to see whether the perceived value by customers influences the continuance intention in the company, that means use the goods and services of the company in future.”

Hypothesis H1. Perceived value by customers has a positive influence on their continuance intention in the company”

….“Because of this, the objective of hypothesis 2 is to see if perceived value on business by consumer has a positive relationship with customer satisfaction.”

“Hypothesis H2. Perceived value has a positive influence on satisfaction of customers”

….”For this reason, the objective of hypothesis 3 is to check if perceived value on business by consumer has a positive relationship with customer trust generated by the company.”

Page 6: “Hypothesis H3. Perceived value has a positive influence on trust generated by the company in the customer”

…“From here, the objective of hypothesis 4 is to see if the satisfaction of the company's customers influences their continuance intention purchase of products and services from that company.”

“Hypothesis H4. Satisfaction of the customers has a positive influence on continuance intention”

…”For this, the objective of hypothesis 5 is to see if the satisfaction achieved by customers is linked with the environmental sustainability.”

“Hypothesis H5. Satisfaction of the customers generated by the company has a positive influence on environmental sustainability”

…”Because of this, the objective of hypothesis 6 is analyze if the satisfaction achieved by customers is related with trust, that means that if the customers are satisfied, that generates them the trust on that company and its actions.”

“Hypothesis H6. Satisfaction of the customers generated by the company has a positive influence on trust”

Page 7: …”From here, the objective of hypothesis 7 is to see whether customer srust is positively related to the intention to continue being the customer or user of this company.”

“Hypothesis H7. Customer trust has a positive influence on their continuance intention in the company”

…“For this reason, the objective of hypothesis 8 is to see whether customer trust is positively related to the environmental sustainability, this means that if customer trusts on the company and its actions, this will have an impact on sustainability.”

Hypothesis H8. Customer trust has a positive influence on environmental sustainability

…”From here, the objective of hypothesis 9 is to see if there is a positive correlation between customer’s intention to continue use goods and services of the company and environmental sustainability.”

Hypothesis H9. Customer’s continuance intention in the company has a positive influence on environmental sustainability

A case study is developed (but is so local) and it is focused on the peculiarities on the one specific country, Spain. It would be interesting analyse if cultural differences can affect the results.

The Authors

We agree with the comments mentioned above. Thus, following the reviewer’s advice, we have put in the conclusion section the following clarification:

Page 17: “Although, this implications cannot be generalized, since previous studies have shown that there are different cultural differences that influence on customers’ behaviors, motivation, emotion or evaluation choice emong others [138]–[140], this means that the result can vary depending on the country.”

The methodology used can be deemed appropriate, and the authors’ presentation of the results is clear and concise, thus facilitating the reader’s understanding, but it would be necessary to deepen in the limitations of the chosen methodology. It would be interesting to know whether the results presented differ from previous studies. This would add value to the findings.

The Authors

We appreciate the comment. Based on the reviewer comment the following clarification has been added.

Page 17: …” As in all studies, this paper is not short of limitations. Our main limitations are in the size, sample and number of studies consulted previously. Another of the limitations could be the methodology used for this study. According to Hair et al. [110], not all are the advantages when use PLS-SEM technique. One of the disadvantage to use PLS-SEM is that it focus on maximizing partial model structures, that means that first it optimizes measurement model parameters and then, in a second step, estimates the path coefficients in the structural model. Another limitation to use PLS‑SEM for theory testing and confirmation is that there is no adequate global measure of goodness of model fit. And finally, these researchers point out that PLS‑SEM parameter estimates are not optimal regarding bias and consistency. Despite the limitations, our study can be considered a first step in the study of restaurant social media on environmental sustainability.”

Finally, in the discussion and conclusion section it would be interesting deep in managerial implications and future research lines.

The Authors

We completely agree with the reviewer on this point. Due to the length limit in this journal, we did not extend the discussion of our results properly, as we should have done. In this revised version, we have considered this concern including a more detailed discussion of managerial implications and adding our future lines of research of our study. Therefore, the following remarks have been added:

Page 17: “This study has some main implications for food and beverage companies managers: first, managers of these companies must pay more attention to the values that pass on to their customers, since there is a positive relationship between customers perceived value from the company and the variables like continuance intention, trust and satisfaction. So, if companies have customers who trust them, are satisfied with the service and goods and plan to continue buying products and services from this company that will lower the possibility that the company will have problems in the future. Secondly, if managers of food and beverage companies want to carry out environmental sustainability campaigns through the social media of restaurants it is important that enhance the confidence of followers towards the restaurant as this will have a decisive influence on the success of the campaign. Finally, managers of food and beverage companies should know that to be sustainable is the duty of all citizens and if they participate to make their consumers aware of it, they will participate and protect environment for future generations.”

“As in all studies, this paper is not short of limitations. Our main limitations are in the size, sample and number of studies consulted previously. In fact, our study can be considered a first step in the study of restaurant social media on environmental sustainability”.

“Related to future research on this topic, it would be interesting to analyse the impact of social networks on environmental sustainability using the sample collected in all regions of Spain. Also compare the result of Spain in one or several countries taking into account the variables of cultural differences. In future studies, we will try to study different sectors, not only food and beverage sector, as well as modifying the background variables of environmental sustainability. In addition, it would be interesting to use different methodology such as the Q Methodology. With this tool we could quantify qualitative answers for customers and will allow us to detect blind spots.”

Reviewer 3 Report

The authors have investigated 9 hypotheses but the theoretical background of selecting all variables such as perceived value, continuity intention, trust and satisfaction, Intention and environmental sustainability is very weak. The is no justification from the theory why only these factors are considered.  It is not clear if the constructs are adopted or new. Some factors have just one/two items.  Other comments include All the numbers should be in point value, a comma is confusing.  In-text citations throughout the paper are not consistent. Such as line 159 (Ulaga, & Chacour, 2001). Lines 173. 195, 197

Author Response

Dear Reviewer 3,

We are pleased to resubmit for publication the revised version of the manuscript “The impact of restaurant social media on environmental sustainability: An empirical study”. First, we would like to thank you for the opportunity to revise and to resubmit our work.

We have carefully taken your comments and insights on board doing our best in the revised version. Therefore, below we will respond to your comments, we also attach the new version of our study, indicating in pale blue the modifications made.

The authors have investigated 9 hypotheses but the theoretical background of selecting all variables such as perceived value, continuity intention, trust and satisfaction, Intention and environmental sustainability is very weak. The is no justification from the theory why only these factors are considered. It is not clear if the constructs are adopted or new. Some factors have just one/two items.

The Authors

We completely agree with the reviewer on this point. Due to the length limit in this journal, we did not extend the discussion of our variables, as we should have done. In this revised version, we have considered this concern including a more detailed discussion of the variables in the section 3. Hypothesis.

Pages 4-5: …“These four variables have been studied in depth separately [54]–[58] and jointly by several researchers [7], [59]–[63]. The conclusions of these studies indicated the importance of these variables for the growth of company’s productivity, financial performance, solvency or short and long term risk reduction. That means that if the companies want to be stable and increase the productive capacity or profitability they have to have the customers that trust them and are satisfied with the products/services/politics of the company. The study carried out for [57] study considers a satisfied customer as an economic asset for the company with high returns and low risk. For all that, we can say that for companies their customers satisfaction, continuance intention, trust and perceived value are an important factors. At the same time consumers are increasingly demanding with the quality of products and services and want to know more about the business and political actions of the company [64]–[67]. Studies along these lines have shown that when consumers choose, they give preference to companies that respect environmental sustainability [68], [69]. All this indicates that there can be a positive and direct relationship between the variables described above and environmental sustainability. Each of these variables and their possible relationships are studied below.”…

We also justified the construct adopted for this study. The construct is new, redesigned for this study, but it is based on the study of Chen and Lin (2015). These authors use several factors with one or more items. Being the study based on the previous research we have not changed number of items for factors. Therefore, the following remarks have been added:

Page 8: …“proposed research model for this study is a new model, since there are no researchs that study the relationship of variables such as costumer trust, satisfaction and continuance intention with environmental sustainability. An empirical study on customer experience and perceived value on sustainable social relationship in blogs using the same variables was carried out by Chen and Lin [7]. Therefore the factors and items of this study have been adapted to our research model.”

Other comments include All the numbers should be in point value, a comma is confusing. In-text citations throughout the paper are not consistent. Such as line 159 (Ulaga, & Chacour, 2001). Lines 173. 195, 197.

The Authors

Thank for comment, we are sorry about the mistake. We have checked and modified the errors.

Round 2

Reviewer 3 Report

The authors have tried to address previous concerns. In addition, I would recommend the following to strengthen the analysis. Overall, PLS is well explained however why all factors have modelled as reflective and none formative needs some justification. 

In the same Sustainability journal, I found one study explaining reflective and formative constructs.  

Sohaib, O.; Kang, K.; Nurunnabi. M, Gender-Based iTrust in E-Commerce: The Moderating Role of Cognitive Innovativeness. Sustainability 201911, 175.

In addition, the authors can further strengthen the discussion and implications of the study by application predictive performance. Such as some of the related papers I found below. I believe the authors should apply IPMA to elaborate on the implications for managerial actions. 

C.M. Ringle, M. Sarstedt, Gain more insight from your PLS-SEM results: The importance performance map analysis. Ind. Manag. Data Syst. 2016116, 1865–1886

O. Sohaib,  K. Kang, and  Miliszewska, I. Uncertainty Avoidance and Consumer Cognitive Innovativeness in E-Commerce. Journal of Global Information Management (JGIM), 27(2), 59-77. 2019. doi:10.4018/JGIM.2019040104

G. Shmueli, S. Ray, J.M. Velasquez Estrada, and S. B. Chatla, "The elephant in the room: Predictive performance of PLS models", Journal of Business Research, Vol. 69 No. 10, 2016, pp. 4552-4564.

Author Response

Dear Reviewer 3,

First, we would like to thank you again for the opportunity to revise and resubmit our paper. We really appreciate the opportunity to publish in Sustainability Journal.

Second, we have sent an updated version of the paper through the online submission site. The most significant changes in the manuscript have been highlighted in pale blue to facilitate your job. Regarding the insights, we have carefully considered them and have incorporated the advice to the best of our ability.

We addressed all of the comments raised by you and present our answers after each of them in blue italics. We hope that all of our comments satisfy your concerns about the paper and it can be published.

The authors have tried to address previous concerns. In addition, I would recommend the following to strengthen the analysis. Overall, PLS is well explained however why all factors have modelled as reflective and none formative needs some justification. In the same Sustainability journal, I found one study explaining reflective and formative constructs.

Sohaib, O.; Kang, K.; Nurunnabi. M, Gender-Based iTrust in E-Commerce: The Moderating Role of Cognitive Innovativeness. Sustainability 2019, 11, 175.

The Authors

We are grateful for your comment. Each variable and the modeling of its factors has been previously studied and analyzed by means of the articles that we include in the bibliographic review. In the work there is a variable with formative character in the second order model that is "Trust". However, we will explain in the article about the modeling. We will take into account the bibliography that you provide us to make the justification.

Page 10: …”In the research model, it can be seen how most variables are modeled with reflective indicators. Although, only "Trust" are modeled with formative indicators when it groups its items in dimensions. The reason for modeling most variables with reflective character was because they were appreciated as an effect of latent variables [113], that is to say, that the reflective indicators are interchangeable [103]. An increase in the variable is reflected in the increase in the indicators since they are highly correlated. With respect to the “Trust” variable that can be found in the second-order model, it has its indicators with formative character because it is a multidimensional construction variable. The formative constructions do not have interchangeable indicators, that is to say, a change in one indicator does not necessarily denote a change in other indicators [113].

In addition, the authors can further strengthen the discussion and implications of the study by application predictive performance. Such as some of the related papers I found below. I believe the authors should apply IPMA to elaborate on the implications for managerial actions.

C.M. Ringle, M. Sarstedt, Gain more insight from your PLS-SEM results: The importance performance map analysis. Ind. Manag. Data Syst. 2016, 116, 1865–1886

Sohaib, K. Kang, and Miliszewska, I. Uncertainty Avoidance and Consumer Cognitive Innovativeness in E-Commerce. Journal of Global Information Management (JGIM), 27(2), 59-77. 2019. doi:10.4018/JGIM.2019040104 Shmueli, S. Ray, J.M. Velasquez Estrada, and S. B. Chatla, "The elephant in the room: Predictive performance of PLS models", Journal of Business Research, Vol. 69 No. 10, 2016, pp. 4552-4564.

The Authors

We would like to thank you for the bibliographical contributions that have helped us to enrich this article, as well as to broaden our knowledge about the different possibilities in analysis of the Smart-PLS software. With regard to the IPMA analysis, it has been added to the article as it advises us.

Pages 15-16: …”6. Importance-Performance Map Analysis

Finally, once the proposed model has been analysed, the importance-performance map analysis (IPMA) is carried out. This analysis can enrich the PLS-SEM analysis, obtaining additional results and findings. This study not only considers the route coefficients, but also considers the average value of the latent variables and their indicators, that means the performance dimension [137], [138]. These novel findings through IPMA provide additional conclusions for management actions as it combines the importance analysis and dimensions of performance in practical PLS-SEM applications [137]. The analysis allows to identify the most important areas of specific actions [139]. The results obtained are represented graphically by contrasting the total effects on the x-axis with the latent variable scores, reflected on a scale from 0 to 100, on the y-axis. The greater the factor yield, the closer the factor is to 100, and all total effects greater than 0.10 are significant at the "p" level less than or equal to 0.05 [137]–[139]. Once the target variable (Environmental Sustainability) has been defined, the results are obtained.

Table 7. IPMA result of the target construct Environmental Sustainability

Constructs

Importance

Performance

Perceived Value

0.360

53.242

Satisfaction

0.109

61.077

Trust

0.419

59.775

Continuance Intention

0.101

46.846

Figure 3. IPMA of the target construct Environmental Sustainability

Once the IPMA has been calculated with respect to Environmental Sustainability in table 7 and figure 3, the yields and effects of the study variables can be appreciated. With respect to the importance of constructs versus the target variable it can be seen how trust has a relatively high importance followed by perceived value, the other two variables have a lower importance. However, it can be seen that the performance of all four variables has very close values, being in the middle approximately, which indicates that they have possibilities for improvement.